# Evaluation of Machine Learning and Traditional Statistical Models to Assess the Value of Stroke Genetic Liability for Prediction of Risk of Stroke Within the UK Biobank

**DOI:** 10.3390/healthcare13091003

**Published:** 2025-04-26

**Authors:** Gideon MacCarthy, Raha Pazoki

**Affiliations:** 1Cardiovascular and Metabolic Research Group, Department of Biosciences, College of Health, Medicine, and Life Sciences, Brunel University of London, Uxbridge UB8 3PH, UK; gideon.maccarthy@brunel.ac.uk; 2Department of Epidemiology and Biostatistics, School of Public Health, Imperial College London, London W2 1PG, UK

**Keywords:** the receiver operation characteristic (ROC), area under the curve (AUC), brier score (BS), integrated calibration index (ICI)

## Abstract

**Background and Objective:** Stroke is one of the leading causes of mortality and long-term disability in adults over 18 years of age globally, and its increasing incidence has become a global public health concern. Accurate stroke prediction is highly valuable for early intervention and treatment. There is a scarcity of studies evaluating the prediction value of genetic liability in the prediction of the risk of stroke. **Materials and Methods:** Our study involved 243,339 participants of European ancestry from the UK Biobank. We created stroke genetic liability using data from MEGASTROKE genome-wide association studies (GWASs). In our study, we built four predictive models with and without stroke genetic liability in the training set, namely a Cox proportional hazard (Coxph) model, gradient boosting model (GBM), decision tree (DT), and random forest (RF), to estimate time-to-event risk for stroke. We then assessed their performances in the testing set. **Results:** Each unit (standard deviation) increase in genetic liability increases the risk of incident stroke by 7% (HR = 1.07, 95% CI = 1.02, 1.12, *p*-value = 0.0030). The risk of stroke was greater in the higher genetic liability group, demonstrated by a 14% increased risk (HR = 1.14, 95% CI = 1.02, 1.27, *p*-value = 0.02) compared with the low genetic liability group. The Coxph model including genetic liability was the best-performing model for stroke prediction achieving an AUC of 69.54 (95% CI = 67.40, 71.68), NRI of 0.202 (95% CI = 0.12, 0.28; *p*-value = 0.000) and IDI of 1.0 × 10^−4^ (95% CI = 0.000, 3.0 × 10^−4^; *p*-value = 0.13) compared with the Cox model without genetic liability. **Conclusions:** Incorporating genetic liability in prediction models slightly improved prediction models of stroke beyond conventional risk factors.

## 1. Introduction

Stroke is one of the leading causes of mortality and long-term disability in adults over 18 years of age globally [1,2], with a detrimental impact on the economy and the cost of healthcare and social services throughout the world. Stroke survivors have a considerably higher risk of mortality when compared with non-stroke patients, not only attributed to the initial stroke but also to stroke-associated consequences and increased cardiac incidence in years after a stroke [3,4,5,6]. Every year, more than 100,000 people in the United Kingdom (UK) suffer from a stroke, and over 1.2 million stroke survivors live in the UK. Stroke incidence and prevalence in the UK are expected to increase by 60% and 120% annually between 2015 and 2035, respectively [7].

Studies have shown that both genetic and non-genetic factors play a critical role in the complex process of stroke events [8]. Stroke risk increases with age, with an estimated 10-year stroke risk in those aged 55 and over. The risk varies by gender and the increasing co-occurrence of risk factors, such as hypertension, diabetes mellitus, atrial fibrillation, high blood cholesterol and lipids, cigarette smoking, physical inactivity, chronic kidney disease, and family history [9].

Twin and family history studies provided early evidence that genetics had a role in stroke risk [10]. Genome-wide association studies (GWASs) have provided further evidence to confirm the role of genetic factors in the occurrence of stroke. More recently, large-scale GWASs, such as the International Stroke Genetics Consortium (ISGC), have identified genetic loci associated with stroke. The MEGASTROKE project identified over 32 loci contributing to stroke risk, revealing the causal role of specific genes and gene regions in stroke origins [11,12]. As a result, greater insight into the genetic indicators of stroke has allowed an opportunity for a deeper evaluation of an individual’s stroke risk, as well as potentially more informed medical and lifestyle decisions that may be preventative measures to reduce the risk of stroke occurrence.

Prediction tools for stroke, such as the Framingham Stroke Risk Profile (FSRP), the American Heart Association (AHA), and the American Stroke Association (ASA), are critical in identifying at-risk individuals early on, allowing for timely treatments and improving outcomes [13,14]. Their developments extend beyond individual treatment, including healthcare policy, budget allocation, and ethical issues for patient data. Advances in artificial intelligence and machine learning are pushing the boundaries of prediction tools, making them more accurate and adaptive to diverse groups of patients [15].

The stroke prediction tools (FRSPs and ASA), as well as the current clinical guidelines for cardiovascular disease prevention, do not evaluate or integrate genetic liability into the risk assessment [16]. Genetic prediction of stroke has the potential to transform stroke prevention and treatment. It has the potential to identify individuals who are at risk of or predisposed to stroke even before clinical symptoms appear. This allows for early treatments, such as lifestyle adjustments or personalized drug programs [17,18,19,20,21]. Genetic polymorphisms in genes associated with stroke or its risk factors have been investigated in stroke risk. Several studies reported a significant association with stroke risk and a genetic liability derived from a set of single nucleotide polymorphisms (SNPs) that were previously identified to have a strong association with stroke or stroke risk factors [22,23,24,25,26,27,28]. Genome-wide genetic liabilities, derived from the combined effects of several genetic variants across the genome, regardless of the strength of their association, have been increasingly tested in the last decades for their effect in health and disease, and previous studies have shown that higher scores of genome-wide genetic liabilities enhance the stroke risk prediction [29,30].

Machine learning models are increasingly applied to predict the risk of complex diseases [31,32,33,34,35,36,37,38,39,40,41,42]. Studies focusing on the prediction of the risk of stroke [32,41,42] have shown that machine learning models outperformed traditional statistical techniques, such as the Cox proportional hazards model. However, there is no consistency on which machine learning model is a better fit. Chen et al. [42] identified artificial neural networks (ANNs), whilst Chun et al. [32] found that gradient-boosted trees (GBTs) were superior to other machine learning models. In addition, Wang et al. [41] identified that the random forest approach outperformed the Cox proportional hazards model.

The predictive value of the genetic factors used in machine learning models is unclear. In a case–control study focusing on patients with atrial fibrillation, Papadopoulou et al. [40] showed that out of multiple machine learning models incorporating a genetic liability, XGBoost outperformed a widely used existing clinical prediction model (CHA2DS2-VASc). The study by Papadopoulou et al. [40] did not include incident stroke, and they created their genetic liability using a selected list of SNPs associated with ischemic stroke (Appendix A).

To our knowledge, there is currently no study in the European general population that provides a comprehensive insight into the prediction of the risk of incident stroke in various scenarios, incorporating machine learning and a stroke genome-wide genetic liability. To fill this gap, our research focused on incorporating a genome-wide genetic liability into machine learning for the prediction of the risk of incident stroke using survival data. This would offer a better understanding of the additional benefit of genetic liability in stroke risk prediction, as well as of how machine learning algorithms perform in comparison to traditional survival models in this context.

We have three main objectives, including (1) assessing the association of whole-genome liability and the risk of future stroke occurrence (incident stroke), (2) assessing the predictive value of stroke genetic liability in the prediction of stroke, and (3) comparing the performance of the Cox proportional hazard model and machine learning models before and after incorporating genome-wide stroke genetic liability into the model.

## 2. Material and Method 

### 2.1. Ethical Approval

The Northwest Multi-Centre Research Ethics Committee approved the UK Biobank (UKB) as a research tissue bank, and all participants involved in the UKB project provided informed consent. The current study is based on UKB data, with the application number 60549. In addition, Brunel University of London’s College of Health, Medicine and Life Sciences Research Ethical Committee approved the use of UKB secondary data (reference 27684-LR-Jan/2021-29901-1).

### 2.2. Study Population

The UK Biobank (UKB) is a prospective observational research study including more than 500,000 adults aged between 40 and 69 years. From 2006 to 2010, participants were recruited from 22 centres across the United Kingdom. The comprehensive description of the UK Biobank study, the acquired data, and a summary of its characteristics are publicly accessible on the UK Biobank website (www.biobank.ac.uk, viewed on 20 June 2021) and in other sources, including Sudlow et al. [43]. During the recruitment stage, detailed information about socioeconomics, demographics, health status, family history of diseases, and lifestyle variables was obtained from the participants through questionnaires and interviews. Several physical measurements were obtained, including height, weight, body mass index (BMI), waist–hip ratio (WHR), systolic blood pressure (SBP), and diastolic blood pressure. The records of UKB study participants were linked to health episode statistics (HES) data and national death and cancer registries.

The current study focuses on a sample of unrelated participants of European ancestry (N = 243,399; Figure 1). In brief, we employed 40 genetic principal components developed centrally by the UKB and used the k-means clustering technique on 502,219 UKB participants to identify persons of European ancestry who had available genetic data (N = 459,042). The study eliminated participants who had withdrawn their informed consent (N = 61), pregnant women, and those who were uncertain about their pregnancy status (N = 278). We excluded participants whose self-reported sex did not match their genetic sex (n = 320). We excluded people who were first and second-degree relatives (N = 33,369) by using a kinship cutoff of 0.0884 for third-degree relatives. We removed individuals (N = 25,340) who had been diagnosed with vascular or cardiac issues by a clinician before or during recruitment. This was carried out to minimize possible confounding, the influence of reverse causality, and selection biases. Participants who used cholesterol-lowering medicine (N = 34,243), quit smoking or drinking due to health reasons or doctor’s advice (n = 58,752), or had missing data on confounders (N = 61,961) were also removed from the dataset.

We subsequently excluded participants who had prevalent stroke cases (N = 248), and self-reported stroke (N = 130). We then merged the data with genetic liability profile data (N = 425,054) calculated for participants with available genotype data (N = 459,042), leaving a final 243,399 unrelated individuals of European ancestry.

**Figure 1 healthcare-13-01003-f001:**
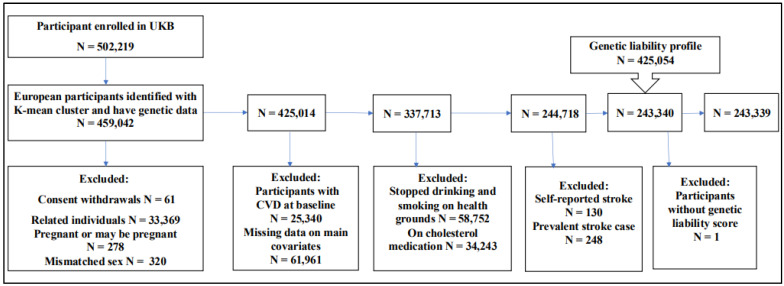
Exclusion criteria of the study: The flowchart for selecting research participants. At the start of this study, the UK Biobank (UKB) had over 500,000 participants. We employed the K-means cluster approach to extract 459,042 European-ancestry subjects. The final dataset had 243,339 people who satisfied the inclusion criteria.

### 2.3. Genotyping and Imputation

The UKB conducted all DNA extraction, genotyping, and imputation. Details of procedures are discussed elsewhere [44,45,46]. To summarize, blood samples from participants were taken at UKB assessment centers, and DNA was extracted and genotyped using the UKB Axiom array. UKB used the IMPUTE4 program [47] to perform the genotyping imputation. The three reference panels used for imputation were the Haplotype Reference Consortium, UK10K, and 1000 Genomes Phase 3. The UKB generated genetic principal components and kinship coefficients centrally to identify related individuals and adjust for population stratification [44,46]. 

### 2.4. Definition of the Outcome 

Our primary outcome in the current study was stroke events, defined according to the International Classification of Diseases 10th revision (ICD-10, I60–I67). In this study, incident stroke was characterized using cerebrovascular disorders ICD-10 code (I600–I609, I610–I619, I630–I639, I64, I650–I659, I660–I669, and I670–I679) for the first stroke event. The current study’s follow-up period is computed from the date of health assessment upon enrolment to the end of March 2017. The participants who did not experience the outcome at the end of the follow-up period were censored.

### 2.5. Demographics and Clinical and Lifestyle Features 

In this study, the conventional risk factors, including age, sex, BMI, diabetes mellitus (DM), hypertension, total cholesterol (TC), low-density lipoprotein (LDL), smoking, and drinking, were considered in all the analyses. A doctor’s diagnosis of diabetes, the usage of insulin, a blood hemoglobin (HbA1c) level greater than or equal to 48 mmol/mol (6.5%), or a glucose level greater than or equal to 7.0 mmol/dL were all considered indicators of diabetes mellitus (DM) [48]. Hypertension is defined as (1) having a recorded SBP greater than or equal to 140 mmHg or DBP greater than or equal to 90 mmHg, (2) having a doctor-diagnosed case of hypertension, or (3) having a record of taking blood pressure (BP)-lowering medication at baseline [49,50]. 

In the UKB, a manual sphygmomanometer or a standard automated device was used to collect two blood pressure readings, separated by a few minutes (https://biobank.ctsu.ox.ac.uk/ukb/ukb/docs/Bloodpressure.pdf (accessed on 22 November 2021)). Using two automatic or two manual blood pressure readings, we calculated the mean SBP and mean DBP. The average of the two values was used for people who had one manual and one automated blood pressure reading. For participants having a single blood pressure record, that one blood pressure reading was used for those participants. For participants using blood pressure-lowering drugs, we increased SBP by 15 mmHg and DBP by 10 mmHg [51]. We excluded individuals with incomplete blood pressure readings from the study. The UKB used a self-reported questionnaire to collect data on participant smoking and alcohol consumption, and categorized respondents as never, previous, and current consumers.

### 2.6. Computation of Genetic Liabilities 

#### Selection of Genetic Variants

We selected a list of genetic variations, in the form of SNPs (Appendix A), that were previously identified in the European population as being associated with stroke [11]. The effect sizes for these SNPs (Appendix A) were derived from GWAS summary statistics data that were published and made publicly available on the GWAS Catalog website (https://www.ebi.ac.uk/gwas/, visited on 12 July 2021). SNPs with a minor allele frequency (MAF) of less than or equal to 0.01 and duplicate, non-biallelic SNPs were not included in the genetic liability calculation for this study. We also conducted an LD pruning technique to exclude SNPs that were in linkage disequilibrium (LD) with one another. When the correlation between SNPs occurs more frequently than expected in a random sample, the SNPs are said to be in LD [52]. LD between two loci is statistically determined by using metrics, such as the correlation coefficient (r^2^) value. This value measures how well the alleles at the two loci correlate with one another. LD pruning removes highly correlated SNPs to avoid the statistical bias and computational inefficiency caused by LD. For this LD pruning process, all pairs of SNPs within a given moving window are evaluated to determine their pairwise LD based on r^2^ value. If any pair of SNPs within the window has an LD larger than the stated threshold, the first SNP will be pruned [53]. The pruning process was implemented in PLINK version 1.9 [54] with the function and parameters “*--indep-pairwise window size = 250 step size = 50 r^2^ = 0.1*”. After the LD pruning procedure, 252,903 SNPs were retained for calculating the genetic liability for stroke based on the Purchell method [54] (Appendix A).

The calculation of genetic liability for stroke was implemented in PLINK version 1.9 with the function “*-score*”. PLINK employs a weighted technique in which the effect size (beta coefficient) of each SNP is used as a weight and is multiplied by the number of risk alleles carried by the participant. The result is then summed up across all SNPs in the calculation of genetic liability.

### 2.7. Data Preprocessing

We preprocessed the dataset by standardizing all quantitative variables, including age, BMI, TC, LDL, and genetic liability using the “*scale*” function in the R package. Categorical variables included sex (male and female), smoking status (never, previous, current), alcohol consumption status (never, previous, current), DM (no, yes), and hypertension (no, yes). Genetic liability was additionally categorized as low, medium, and high risk according to its tertiles to ease the analysis per subgroup of genetic liability.

## 3. Statistical Analysis 

For a statistical description of the baseline characteristics of our study population, we used the “*gtsummary*” and “*table1*” packages in the R-program Windows version 4.4.1 for statistical analyses [55]. The categorical variables were summarized using frequencies and percentages, and the numerical variables were expressed as the mean (SD). The chi-square test was used to compare differences in binary outcome (stroke event and non-event) in relation to categorical variables. For continuous variables, the Wilcoxon rank sum test was used. We used the “*cor*” function to calculate the correlation matrix and the Pearson correlation between variables and the “*ggcorrplot*” function from the *ggcorrplot* package to visualize the correlation matrix. We then examined the correlation matrix using the “*findCorrelation*” function from the *caret* package to identify highly correlated features. In this study, we set the Pearson correlation (r^2^ = 0.8) as the threshold for collinearity [56,57].

The feature selection procedure began with (1) selecting risk factors known to be associated with stroke and (2) were associated with stroke in our data using the univariate Cox regression (*p*-value less than 0.05 for inclusion). (3) We then used the correlation coefficient to assess the correlation among the selected risk factors (r^2^ less than 0.8 for inclusion). The finally selected risk factors were used to construct the conventional risk factor model (model 1).

### 3.1. The Relationship Between Genetic Liability and Stroke

We used univariable and multivariable Cox proportional hazard regression to assess the relationship between stroke genetic liability (continuous and categorical) and the risk of incident stroke over the follow-up period. Hazard ratios (HRs) are commonly used to evaluate outcomes, such as survival time and time to event. HR is a measure used in survival analysis to compare the risk of an event occurring at any given point in time between two groups.

Following the univariable Cox proportional hazard regression analysis (model 1, unadjusted), three multivariable adjustment Cox proportional hazard regression models (models 2, 3, and 4) were developed to examine the potential influence of known cardiovascular risk factors on the relationship between genetic liability and stroke risk. In model 2, we adjusted for age and sex. In model 3, BMI, hypertension, DM, and LDL were adjusted in addition to age and sex, and in model 4, we further adjusted for drinking status and smoking status (the full model). We identified statistical significance when the associations established a two-sided *p*-value less than 0.05. We assessed the proportional hazard (PH) assumptions using statistical testing (the “*cox.zph*” function) and a visual examination of scaled Schoenfeld residuals (the “*ggcoxzph*” function) using the R *survival* package version 3.8-3.

### 3.2. Prediction Models Development

In this study, two sets of prediction models were created for each technique to predict the incidence of stroke. These were (1) the conventional risk factors model (the model without genetic liability), which combines the conventional risk factors selected from univariable association tests, and (2) the integrated prediction model, which combines the conventional risk factors with genetic liability for stroke (genetic risk). The input variables and output in the current study are displayed in Appendix A.

Using the “*createDataPartition*” function from the *caret* package, we randomly partitioned our dataset into a training set (70%; N = 170,381; event = 1382; non-event = 168,999) and a testing set (30%; N = 73,018; event = 591; non-event = 72,427).

To predict the risk of incident stroke, we used the training data to create prediction models using the Cox proportional hazard. Cox proportional hazard regression [58,59] is a popular statistical approach for assessing survival data and determining the association between the time until an event (such as death, failure, or illness recurrence) occurs and one or more predictors. We implemented the Cox proportional hazard models using the “*coxph*” function from the *Survival* package in R software version 3.8-3.

In addition, we developed three machine learning techniques in the training set, including the gradient boosting machine (GBM) models, decision tree (DT), and random forest (RF), to predict the risk of stroke. We then assessed the performance of each model in the testing set (Appendix A).

The decision tree is one of the common and simple methods used for classification and regression applications. It works by dividing a dataset into smaller subgroups depending on feature values and then generating a decision tree [60]. The decision tree method in this study was implemented using the “*rpart*” function from the recursive partitioning and regression trees (*rpart*) package, and the minimum number of observations required to split a node at each branch was set to 4. The complexity parameter (cp) to control the size of the decision tree and prevent overfitting was set at 0.001, meaning that a split must improve the model’s fit by at least 0.1% to be considered. This parameter is used to save computing time by removing irrelevant splits. The optimal decision tree was obtained with the “*prune*” function. The function removes the trees that do not meet the complexity parameter value. That is, the “*prune*” function removes branches without a lack of fit reduction (measured by the residual sum of squares; RSS) as determined by the complexity parameter value. This process reduces the risk of overfitting the training data.

Random forest is a popular machine learning model for classification and regression. It creates ensembles from decision trees and combines their results to make a final decision [61]. The random forest models were built using the “*ranger*” function from the *ranger* package. The number of trees to be fitted was set to a value of 500. To control the model’s complexity and performance, the number of variables randomly selected at each split when growing the trees was set to a value of 3 (“*mtry*”). This is justified, as the optimal *mtry* value considered for classification models is calculated as the square root of the total number of variables (nine variables in the current study). The value of *mtry* can significantly affect the OOB (out-of-bag) error. The OOB error is an unbiased estimate of the prediction error calculated by using samples not included in the bootstrap sample for a given tree. It serves as a cross-validation mechanism that is integrated into the random forest. A smaller *mtry* value increases the randomness and diversity among the trees, which can help reduce overfitting and potentially lower the OOB error. However, if the *mtry* value is too small, the trees might not capture enough information, leading to higher OOB error. The *mtry* and OOB error are critical in optimizing the random forest model. The range of values for *mtry* was examined by the *ranger* package version 0.17.0, and the *mtry* value that minimizes OBB error was selected as the optimal value in the construction of the random forest model. We additionally built gradient boosting machine models using the *gbm* package version 2.2.2 to predict the risk of stroke. The gradient boosting machine models integrate predictions from many weak learners to increase total prediction accuracy [60]. The number of trees to be fitted was set to a value of 500. The highest number of permissible variable interactions was set to 3. The shrinkage parameter to control the learning rate or step-size reduction was set to a value of 0.01. The parameters of the machine learning models were determined using 10-fold cross-validation (CV). In this study, the parameters with the smallest CV root mean square error (RMSE), CV error (xerror), and OOB error were utilized to develop the GBM, DT, and RF prediction models, respectively.

## 4. Model Performance Assessment

To determine the predictive performance of each prediction model, we used the Platt scaling method [62], also known as the sigmoid method, which is commonly used in machine learning methods for binary data. This method calibrates the output of the prediction models. Platt scaling transforms the output from classification models into a probability distribution. Here, we passed the probability estimates from machine learning models through a trained sigmoid function [62] using univariable logistic regression. In this logistic regression, a variable containing probability estimates for each participant was used as an independent variable. The binary outcome (stroke) served as the dependent variable [63]. The output from this logistic regression provided a new scaled probability estimate that helped calibrate the models. The calibration of a prediction model ensures that the predicted risks are accurate and align with the actual proportions of the event. A prediction model is said to be calibrated if the model’s outcome matches the observed proportions of the event [64]. To assess the agreement between the calibrated probabilities (created using Platt scaling) and the observed patient stroke outcomes, we additionally used the “*pmcalibration*” function from *pmcalibration* in R package. This method allows for nonlinear relationships between the predictors and the response variables. Complementary log–log transformed predicted probabilities were applied to the splines to produce calibration measures for a time-to-event outcome.

The calibration metrics used to assess the model calibration in this study were the Brier score (*BS*) and average absolute difference (*Eavg*), also known as the integrated calibration index (*ICI*). The BS is the mean squared difference between the predicted probabilities and the actual outcomes, and it measures both discrimination and calibration [63]. BS ranges from 0 (perfect prediction and calibration) to 1 (worse prediction and calibration). ICI measures the average absolute deviation between the predicted and observed probabilities, providing an overall assessment of calibration quality [64]. It provides a single, summary measure of calibration quality, making it easier to compare different models or assess changes in calibration over time. An ICI of 0 represents perfect calibration and an ICI of 1 represents worse calibration, suggesting that the predicted probability deviates from the observed events. To calculate ICI and BS, we used the “*pmcalibration*” and “*brier*” functions implemented within the *pmcalibration* and *gmish* packages, respectively.

To assess the discrimination performance of the models, we calculated the area under the curve (AUC) using the *pROC* package in the R program. We reported the AUC, ICI, and BS values of various models. Greater values of AUC and smaller values of ICI and BS indicate improved discrimination and calibration of the model. The overview of the model performance assessment is presented in Appendix A.

### Assessment of the Predictive Value of Genetic Liability

We assessed the predictive value of genetic liability as an additional predictor to the conventional risk factors in each prediction model by estimating the improvement in the AUC, integrated discrimination improvement (IDI), and continuous net reclassification index (NRI). NRI measures the effectiveness of a new model in reclassifying individuals into different risk categories compared to an existing model. At the same time, IDI evaluates the model’s ability to differentiate between cases and non-cases after adding a new variable. It compares the average predicted probability for cases and non-cases in the old and new models [65]. The NRI and IDI were calculated to assess model improvement following the inclusion of genetic liability in the models. This was implemented using the “reclassification” function from the *PredictABEL* package version 1.2-4 in the R-program. Higher IDI value indicated better discrimination, and higher NRI value indicated better risk reclassification by the new model [66,67,68]. The above performance metrics have been discussed in detail, elsewhere [63] and in our previous work [33].

## 5. Results

### 5.1. Study Characteristics

Table 1 presents the baseline characteristics of the study. The study included 243,339 unrelated UK Biobank participants of European ancestry. The average age of participants included in the study was 55.4 (SD = 7.98) years at recruitment. Over half of the sample were women (N = 141,212; 58%). During a median follow-up of 8.22 years, 1973 first-ever stroke episodes, of which 45.3% of patients were women, were recorded among the participants.

In the overall sample, 76,397 participants (31.4%) were current smokers, and 228,349 participants (93.8%) were current alcohol drinkers. All the conventional risk factors included in the analysis showed a statistically significant association with the risk of incident stroke in univariate analysis except total cholesterol (Table 1). The prevalence of DM within the sample was 2.9% (N = 6939) while the prevalence of hypertension was 47.8% among the participants (N = 116,216). The univariable Cox association analysis results indicated that age, sex, BMI, hypertension, DM, LDL, alcohol use, and smoking history were statistically associated with the risk of stroke (Table 1). These variables were used as the features to construct conventional risk factor models. The correlation matrix (Figure 2) between the characteristics in the study demonstrated that total cholesterol and LDL were highly correlated (r^2^ = 0.94). LDL was used in the further analysis and feature selection.

**Table 1 healthcare-13-01003-t001:** Baseline characteristics of the study population stratified for stroke event and non-stroke event within the UK Biobank population.

Characteristic	Overall (N = 243,399)	Non-Event (N = 241,426)	Stroke Event (N = 1973)	HR (95% CI)	*p*-Value
DM, yes; n (%)	6939 (2.9%)	6826 (2.8%)	113 (5.7%)	2.08(1.72, 2.51)	<0.001
Hypertension, yes; n (%)	116,216 (47.7%)	114,840 (47.6%)	1376 (69.7%)	2.52(1.29, 2.78)	<0.001
Sex, male; n (%)	102,187 (42.0%)	101,107 (41.9%)	1080 (54.7%)	1.67 (1.53, 1.83)	<0.001
Age (years), mean (SD)	55.4 (7.98)	55.4 (7.98)	60.0 (7.14)	1.93 (1.83, 2.03)	<0.0001
Body mass index (kg/m^2^), mean (SD)	26.8 (4.57)	26.8 (4.57)	27.4 (4.83)	1.12 (1.08, 1.17)	<0.001
Total cholesterol (mmol/L), mean (SD)	5.91 (1.06)	5.91 (1.06)	5.94 (1.09)	1.03 (0.98, 1.07)	0.30 *
LDL (mmol/L), mean (SD)	4.68 (2.37)	4.67 (2.36)	5.03 (2.51)	1.03 (1.03, 1.12)	0.002
Smoking					
Current; n (%)	76,397 (31.4%)	75,647 (31.3%)	750 (38.0%)	REF	REF
Previous; n (%)	2900 (1.2%)	2855 (1.2%)	45 (2.3%)	1.58 (1.17, 2.13)	0.003
Never; n (%)	164,102 (67.4%)	162,924 67.5%)	1178 (59.7%)	0.73 (0.67, 0.80)	<0.001
Alcohol					
Current; n (%)	228,349 (93.8%)	226,556(93.8%)	1793 (90.9%)	REF	REF
Previous; n (%)	7082 (2.9%)	6996 (2.9%)	86 (4.4%)	1.55 (1.25, 1.93)	<0.001
Never; n (%)	7968 (3.3%)	7874 (3.3%)	94 (4.8%)	1.50 (1.22, 1.85)	<0.001

The *p*-value is from a univariate analysis of the Cox proportional hazard model, comparing the distribution of the baseline characteristics among stroke and non-stroke event. * Not significant; DM = diabetes mellitus; HR = hazard ratio; CI = confidence interval; REF: reference.

**Figure 2 healthcare-13-01003-f002:**
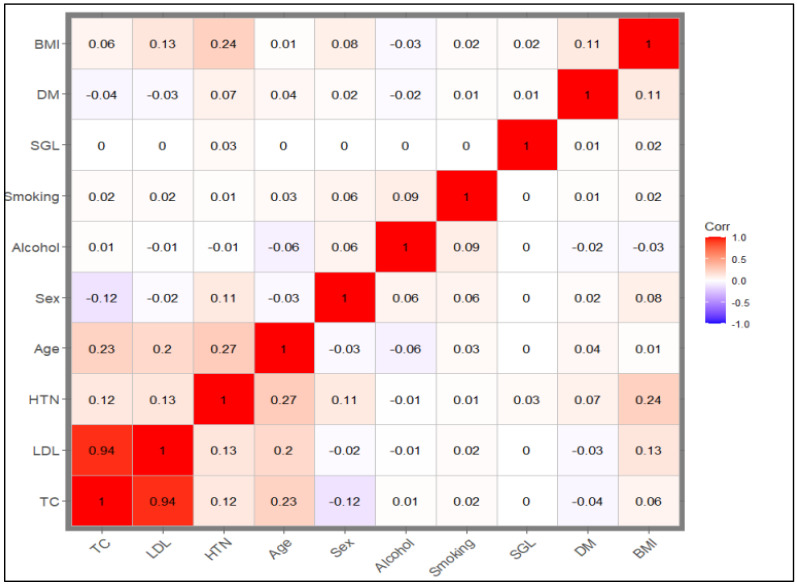
Correlation matrix plot: The plot shows the correlation coefficients between numerical features. TC and LDL are highly correlated (r^2^ > 0.8). TC was excluded from further analysis (prediction model construction). BMI: body mass index; TC: total cholesterol; LDL: low-density lipoprotein cholesterol; HTN: hypertension; SGL: stroke genetic liability.

### 5.2. The Association of Genetic Liability with Incident Stroke

The Kaplan–Meier curve showed differences in stroke incidents and cumulative hazard between the high-risk and low-risk genetic liability groups (Figure 3). Each unit (standard deviation) increase in genetic liability increases the risk of incident stroke by 7% (HR = 1.07, 95% CI = 1.02, 1.12, *p*-value = 0.003; Table 2; Figure 4).

The risk of stroke was greater in the higher genetic liability group, as demonstrated by a 14% increased risk (HR = 1.14, 95% CI = 1.02, 1.27, *p*-value = 0.02) compared with the low genetic liability group. The global Schoenfeld *p*-value from the Schoenfeld test (*p*-value = 0.14; Table 3) indicates that the proportional hazard (PH) assumption is reasonable for the model (Appendix A).

**Table 2 healthcare-13-01003-t002:** The result of the univariable Cox proportional hazard model for the association of genetic liability (categorical and continuous) with incident stroke within the UK Biobank population.

Genetic liability Level	HR (95% CI)	*p*-Value	HR (95% CI)	*p*-Value	HR (95% CI)	*p*-Value	HR (95% CI)	*p*-Value
	Model 1		Model 2		Model 3		Model 4	
Moderate risk	1.06 (0.95, 1.18)	0.31	1.06 (0.95, 1.18)	0.31	1.05 (0.94, 1.17)	0.04	1.05 (0.94, 1.17)	0.40
High risk	1.15 (1.03, 1.28)	0.01	1.16 (1.04, 1.30)	0.01	1.14 (1.02, 1.27)	0.02	1.14 (1.02, 1.27)	0.02
Genetic liability (continuous)	1.08 (1.03, 1.13)	<0.001	1.08 (1.03, 1.13)	<0.001	1.07 (1.03, 1.12)	0.002	1.07 (1.02, 1.12)	0.003

Model 1: univariable Cox proportional hazard. Model 2: adjusted for age and sex. The low genetic risk group was considered as the reference. Model 3: adjusted for age, sex, BMI, LDL, HTN, and DM. Model 4: adjusted for age, sex, BMI, LDL, HTN, DM, smoking status, and alcohol status. BMI: body mass index; LDL: low-density lipoprotein cholesterol; HTN: hypertension; DM: diabetes mellitus.

**Figure 3 healthcare-13-01003-f003:**
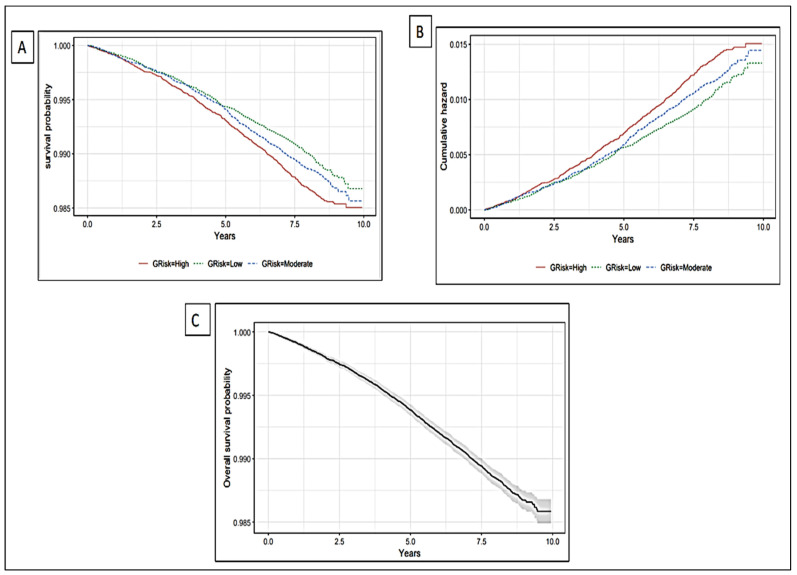
Survival probability and cumulative hazard plot stratified by genetic risk level: (**A**) Survival probability plot stratified by genetic risk level. (**B**) Cumulative hazard plot stratified by genetic risk level. (**C**) Overall survival probability of the study population. (**A**,**B**) demonstrate the difference in the risk of stroke between genetic liability categories. (**C**) illustrates the change in the risk of stroke over time. The grey area surrounding survival probability line in panel (**C**) represents the 95% confidence interval.

**Figure 4 healthcare-13-01003-f004:**
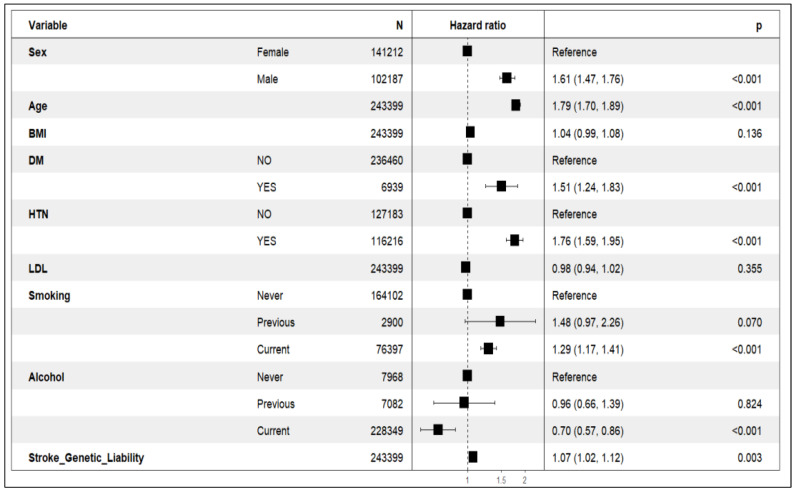
Forest plot of the full Cox proportional hazard model: The vertical line at the hazard ratio (HR) = 1 is the reference line. The horizontal line represents the confidence interval (CI). HTN: hypertension.

### 5.3. Prediction Value of the Conventional Factors

Table 4 summarizes the performance of the prediction models in the testing set. We considered predictions only up to the median follow-up time of 8.22 years.

The Cox proportional hazard model with the conventional risk factors (the model without genetic liability) showed a moderate performance and discrimination (AUC= 69.43; 95% CI = 67.30, 71.56; BS = 0.01, and ICI = 0.002) compared with the gradient boosting machines approach (AUC = 69.34; 95% Cl = 67.23, 71.50; BS = 0.01, and ICI = 0.001), the decision tree models (AUC = 67.58; 95% CI = 65.46, 69.70, BS = 0.01, and ICI = 0.001), and the random forest model, which showed the lowest performance (AUC = 65.62; 95% CI = 65.48, 67.55, BS = 0.01, and ICI = 0.003). The ROC plots of these models are presented in Appendix A. The result from the decision tree model indicates that age, hypertension, and sex are the most relevant predictors of stroke.

**Table 3 healthcare-13-01003-t003:** Assessment of the proportional hazard (PH) assumption using the global Schoenfield test.

Characteristics	Chi-Square	df	*p*-Value
Sex	0.42	1	0.52
Age	0.82	1	0.37
BMI	7.49	1	0.01
DM	0.08	1	0.78
HTN	0.14	1	0.71
LDL	0.36	1	0.55
Smoking	1.40	1	0.24
Alcohol	0.12	1	0.73
SGL	2.48	1	0.12
GLOBAL	13.43	9	0.14

The table illustrates an assessment of the proportional hazard (PH) assumption using the global Schoenfield test. The test indicated a global *p*-value of 0.14, indicating no significant time-dependent joint effect on the covariates. SGL: stroke genetic liability; BMI: body mass index; LDL: low-density lipoprotein cholesterol; DM: diabetes mellitus; HTN: hypertension; df: degree of freedom.

**Table 4 healthcare-13-01003-t004:** The result of the prediction value of the stroke genetic liability score for incident stroke in the UKB.

	Models	AUC 95%CI	NRI (95% CI)	*p*-Value for NRI	IDI (95% CI)	*p*-Value for IDI	Brier Score	ICI
Coxph	Model 1	69.43 (67.30, 71.56)	REF	REF	REF	REF	0.01	0.002
	Model 2	69.54 (67.40, 71.68)	0.20 (0.119, 0.285)	0.00	1.0 × 10^−4^ (0.000, 3.0 × 10^−4^)	0.14	0.01	0.002
GBM	Model 1	69.34 (67.23, 71.50)	REF	REF	REF	REF	0.01	0.001
	Model 2	69.38 (67.26, 71.50)	−0.11 (−0.193, −0.027)	0.01	0.00 (−1.0 × 10^−4^, 1.0 × 10^−4^)	0.61	0.01	0.001
DT **	Model 1	61.40 (59.30, 63.40)	REF	REF	REF	REF	0.01	0.001
	Model 2	61.40 (59.30, 63.40)	0.00 (0.00, 0.00)	NaN	0.00 (0.000, 0.000)	NaN	0.01	0.001
DT	Model 1	67.58 (65.46, 69.70)	REF	REF	REF	REF	0.01	0.001
	Model 2	67.58 (65.46, 69.70)	REF	REF	REF	REF	0.01	0.001
RF	Model 1	65.62 (63.48, 67.75)	REF	REF	REF	REF	0.01	0.003
	Model 2	65.35 (63.18, 67.52)	0.17 (0.087, 0.249)	5.0 × 10^−5^	0.00 (−7.0 × 10^−4^, 8.0 × 10^−4^)	0.98	0.01	0.003

Model 1 (the basic model) features: age, sex, BMI, HTN, DM, LDL, smoking status, and alcohol status. Model 2 features: age, sex, BMI, HTN, DM, LDL, smoking status, alcohol status, and genetic liability. BMI: body mass index; LDL: low-density lipoprotein cholesterol; HTN: hypertension; DM: diabetes mellitus. DT **: decision tree built without pruning parameters. REF: reference; NaN: not a number; NRI: continuous net reclassification index; IDI: integrated discrimination; ICI: integrated calibrated index. ICI is based on a calibration curve estimated for a time-to-event outcome (time = median 8.20 years of follow-up) via a restricted cubic spline using complementary log–log transformed predicted probabilities with the “pmcalibration” function in the R program. In the reclassification analysis, the decision tree (DT) did not remarkably enhance predictions over the baseline (reference). This causes the standard error to be zero, and the NRI and IDI statistics to be near zero, resulting in NaN *p*-values.

### 5.4. Prediction Value of Genetic Liability

The prediction value of the Cox proportional hazards model improved slightly when the stroke genetic liability was incorporated into the model with conventional risk factors (AUC = 69.54; 95% CI = 67.40, 71.68; AUC change = 0.16%; Table 4). We also observed a slight improvement in risk reclassification, leading to an overall NRI value of 0.20 (95% CI = 0.119, 0.285; *p*-value = 0.00; Table 4). The IDI value of the Cox proportional hazard was negligibly improved by 1.0 × 10^−4^ (95% CI = 0.000, 3.0 × 10^−4^; *p*-value = 0.14; Table 4).

The gradient boosting machine model slightly improved in prediction performance (AUC = 69.38; 95% CI = 67.26, 71.50; Table 4) but deteriorated in NRI by a value of −0.11 (95% CI = −0.193, −0.027; *p*-value = 0.01; Table 4) after adding the stroke genetic liability. There was no improvement in the overall IDI value using any of the machine learning models.

Using decision tree (AUC = 67.58, 95% Cl = 65.46, 69.70, BS = 0.01, and ICI = 0.001) or random forest (AUC = 65.35; 95% CI = 65.48, 67.55, BS = 0.01, and ICI = 0.003) models, no improvement in prediction performance was observed adding genetic liability (Table 4). The overall NRI for random forest was improved by NRI = 0.17 (95% CI = 0.087, 0.249; *p*-value = 5.0 × 10^−5^; Table 4) but not for the decision tree technique. The ROC plots of these models are presented in Appendix A.

## 6. Discussion

### 6.1. Main Findings

The present study included genome-wide stroke genetic liability (using 252,903 genetic variants) for 243,399 participants of European descent over a median follow-up of 8.22 years. Our findings indicate that (1) the genome-wide stroke genetic liability is independently associated with the risk of stroke, (2) a prediction model integrating the genome-wide stroke genetic liability provides a slight improvement in prediction performance beyond the conventional risk factor for stroke, and (3) the Cox proportional hazard method showed better prediction performance than machine learning models (random forest, gradient boosting machines, and decision tree) with or without incorporation of genetic liability in the model.

This study’s first finding, i.e., that stroke genome-wide genetic liability increases the risk of stroke, is consistent with previous studies [22,23,24,25,26,29] including studies by Myserlis et al. [22], Rutten-Jacobs et al. [23], Yang et al. [24], Abraham et al. [25], Verbaas et al. [26], and Hachiya [29] that reported that stroke genetic liability is a strong independent predictor of risk of future stroke occurrences.

These previous studies mainly calculated stroke genetic liability based on a limited selection of single-nucleotide polymorphisms (SNPs) that have strong associations with the traits. Our result is a step forward in the sense that we present the risk of stroke imposed by a whole-genome genetic liability of stroke in a European setting. Yang et al. [24] estimated a whole-genome genetic liability of stroke (stroke and its subtypes) in China Kadoorie Biobank and showed that the genetic liability of stroke increases risks of any stroke (14%), ischemic stroke (7%), and intracerebral hemorrhage (10%). We observed a 15% greater risk of any stroke among European participants with a high genome-wide stroke genetic liability compared with those with a low genetic liability which is comparable to the study by Yang et al. [24] in a Chinese population. Our study also differs from previous studies, including the definition or classification of the outcome and sample characteristics. We defined stroke events as any cases of (1) ischemic stroke, (2) intracerebral hemorrhage, (3) subarachnoid hemorrhage, (4) other cerebrovascular disease, or (5) stroke that is not specified as hemorrhage or infarction. Thus, we captured a broader definition of stroke, which could have increased the stroke diversity in our analysis. To investigate the relationship between genetic liability and stroke, Rutten-Jacobs et al. [23] generated a genetic liability from 90 SNPs associated with stroke (at a *p*-value less than 1 × 10^−5^). They demonstrated a 7 to 13% increase in the risk of stroke for each standard deviation increase in genetic liability. Myserlis et al. [22] and Abraham et al. [25] included the genetic liability of stroke within a meta-scoring technique that combined 19–21 distinct genetic liabilities to form a metaGRS. These studies found that the metaGRS was associated with an increased risk of incidence of intracerebral hemorrhage [22] and ischemic stroke [25]. Myserlis et al. showed a 15% increase in the risk of intracerebral hemorrhage and Abraham et al. showed a 26% increase in the risk of ischemic stroke for each standard deviation increase in the metaGRS. The association was stronger than any of the individual genetic liabilities included in the metaGRS. However, the results from Myserlis et al. and Abraham et al. did not distinguish the effect of the genetic liability of stroke per se, as the stroke genetic liability was integrated into a MetaGRS comprising 19–21 distinct genetic liabilities for various traits. Abraham et al. included several genetic liabilities for multiple stroke-related phenotypes, including ischemic stroke, any stroke, small vessel stroke, large artery stroke, cardioembolic stroke, and several stroke risk factors in their metaGRS. Myserlis et al. included genetic liabilities for multiple phonotypes including white matter hemorrhage (n = 87,951 SNPs) and small vessel stroke (n = 2162 SNPs) within the metaGRS calculation. While metaGRS has been found to improve risk prediction, there may be some biases in prediction performance because it was built using elastic-net regression. Additionally, certain SNPs included in the calculation of individual phenotypes’ genetic liabilities may be associated with several phenotypes [26]. Therefore, the metaGRS may contain overlapping information due to possible correlation among the genetic liabilities included in the metaGRS [69]. Our approach to considering genome-wide genetic liability for stroke aimed to capture the polygenic component of stroke, i.e., we had no statistical significance threshold for the selection of SNPs associated with stroke. Thus, we included all SNPs, even those with small or non-significant effects. It is known that this approach would increase the accuracy of the effect estimated for genetic liability and would, therefore, improve accuracy in the identification of high-risk individuals [70,71].

Our prediction models demonstrated that the genome-wide stroke genetic liability may slightly enhance (1) overall stroke prediction performance to distinguish the cases (the Cox proportional hazards model and the gradient boosting machine) and (2) correct classification of individuals at risk beyond conventional risk factors (the Cox proportional hazards model and random forest). However, none of our models demonstrated statistically improved predicted probabilities for cases and non-cases based on IDI.

Our findings from prediction analysis are supported by the results reported in previous studies [40,72,73] which observed that including both genetic liability and conventional risk factors in risk prediction models improves the discrimination performance compared to using only conventional risk factors. Papadopoulou et al. [40] used genetic liability based on 28 SNPs in a European population focused on ischemic stroke in patients with atrial fibrillation (AF). They observed that XGBoost performed better than the CHA2DS2-VASc model, an existing clinical model for calculating stroke risk for patients with atrial fibrillation. Cárcel-Márquez et al. [72] used genetic liability based on 93 SNPs to predict cardioembolic stroke in the European population using logistic regression while Jung et al. [73] used genetic liability based on 16 SNPS to predict stroke in a Korean population using Cox proportional hazard regression. Our best model performed better than the models of Papadopoulou et al. [40] and Jung et al. [73]. Our study population differed from the populations studied by Papadopoulou et al. [40] and Jung et al. [73]. However, the risk values identified in the current study are smaller than those published by Cárcel-Márquez et al. [72]. It should be emphasized that Cárcel-Márquez and colleagues did not use time-to-event data and instead focused on cardioembolic stroke.

Unlike previous studies, which implied that machine learning algorithms outperform traditional statistical approaches in the prediction of stroke [32,41,42], the current study indicated that the Cox proportional hazard regression models outperformed all the machine learning models in the context of time-to-event data for stroke. This could be due to the small number of events (1973 stroke events) and few predictors (up to 9 predictors) in this study. These two reasons are considered as reasons for Cox models to outperform machine learning [74]. We found that genetic liability improved stroke risk classification for less than 1% of the subjects. Health economy studies could consider investigating if using this information in the identification of high-risk individuals to target for stroke prevention programs could make a significant cost-effective change in stroke-related expenses.

The large sample size of UK Biobank and the number of incident strokes enabled the statistical power for our analysis in which we used time-to-event data for over 200,000 individuals of European ancestry, with a median follow-up of 8.22 years. A distinctive feature and the strength of our study compared with previous studies is that we generated genetic liability for stroke using over 250,000 genetic variants.

Validation in external cohort datasets could improve the precision of our findings. To minimize lack of validation in external cohorts, we internally validated our machine learning models in the testing set, where we randomly partitioned the data into a training set (70% of participants) for developing the prediction models and a testing set (30% of participants) to evaluate the prediction models’ performance.

### 6.2. Implication

Genetic predisposition to stroke had minimal impact on improving stroke risk prediction, benefiting approximately one percent of the population. Since genetic liability improved prediction for only a small percentage of the population, its application in clinical practice is uncertain. Conventional risk factors may still have more influence on the prediction of stroke. The findings suggest that genetic liability alone has limited predictive value for most people, but they might still have a role in highly targeted interventions.

In terms of cost effectiveness, given that only a small percentage of the population benefits from genetic risk scores, health economics studies are needed to establish if the costs of genetic testing outweigh the potential improvements in stroke prevention.

## 7. Conclusions

In conclusion, incorporating genetic liability into stroke risk prediction models could slightly improve prediction performance and should be considered when predicting the risk of stroke. Cox proportional hazard models should be given priority over machine learning models in the prediction of the risk of stroke.

## Data Availability

The data used in this study is available on request from the UK Biobank.

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
