# Peer review of "Evaluation of Machine Learning and Traditional Statistical Models to Assess the Value of Stroke Genetic Liability for Prediction of Risk of Stroke Within the UK Biobank"

_healthcare, 2025, doi:10.3390/healthcare13091003_

Round 1
Reviewer 1 Report
Comments and Suggestions for Authors
Create a workflow diagram illustrating the inputs and output for machine learning models.
Justify why machine learning outperformed statistical analysis.
Explain the selection of the following models: Cox Proportional Hazards (CoxPH), Gradient Boosting Model (GBM), Decision Tree (DT), and Random Forest (RF). Clarify why more advanced machine learning models were not used
Provide a detailed conclusion summarizing key findings and their implications.
Avoid using the word "our" in the manuscript, specifically in line 486.
Incorporate previous studies’ results in the table with proper citations to strengthen the discussion.
In table 4, why Random Forest (RF) performed worse than Decision Tree (DT) in this case (RF = 65, DT = 67), despite RF typically outperforming DT.
In Table 4, justify why some p-values are marked as REF or NaN and explain their significance.
Figure 3: Add missing subcaptions and provide a clear description of panels A, B, and C.
Improve readability in Section 4.
Reviewer 2 Report
Comments and Suggestions for Authors
This study investigates the potential added predictive value of incorporating genome-wide stroke genetic liability into traditional and machine learning models for stroke risk prediction. Following major comments:
- Was it appropriate to exclude persons with non-European heritage, and would this have limited how broadly the results could be applied?
- Did preliminary tests or literature references sufficiently support the selection of parameters for machine learning models (e.g., number of trees in RF, learning rate in GBM)? The disease categorization job has a number of papers, including doi: 10.1371/journal.pone.0268555.
- Do the authors explain why, in contrast to the Cox model, introducing genetic liability has no effect on machine learning models?
The English could be improved to more clearly express the research.
Reviewer 3 Report
Comments and Suggestions for Authors
Report is attached

Report is attached
Round 2
Reviewer 1 Report
Comments and Suggestions for Authors
I am pleased to inform you that the authors have incorporated the suggested revisions. Accept in present form
Comments on the Quality of English LanguagePlease check the grammer.
Author Response
Thank you for pointing this out. We have accordingly revised the manuscript.